# Uncharacterized Proteins CxORFx: Subinteractome Analysis and Prognostic Significance in Cancers

**DOI:** 10.3390/ijms241210190

**Published:** 2023-06-15

**Authors:** Pavel Ershov, Evgeniy Yablokov, Yuri Mezentsev, Alexis Ivanov

**Affiliations:** Institute of Biomedical Chemistry, Moscow 119121, Russia; evgeniy.yablokov@ibmc.msk.ru (E.Y.); yu.mezentsev@gmail.com (Y.M.); alexei.ivanov@ibmc.msk.ru (A.I.)

**Keywords:** uncharacterized proteins, open-reading frame, subinteractome, gene expression, cancers, tumors

## Abstract

Functions of about 10% of all the proteins and their associations with diseases are poorly annotated or not annotated at all. Among these proteins, there is a group of uncharacterized chromosome-specific open-reading frame genes (CxORFx) from the ‘Tdark’ category. The aim of the work was to reveal associations of CxORFx gene expression and ORF proteins’ subinteractomes with cancer-driven cellular processes and molecular pathways. We performed systems biology and bioinformatic analysis of 219 differentially expressed CxORFx genes in cancers, an estimation of prognostic significance of novel transcriptomic signatures and analysis of subinteractome composition using several web servers (GEPIA2, KMplotter, ROC-plotter, TIMER, cBioPortal, DepMap, EnrichR, PepPSy, cProSite, WebGestalt, CancerGeneNet, PathwAX II and FunCoup). The subinteractome of each ORF protein was revealed using ten different data sources on physical protein–protein interactions (PPIs) to obtain representative datasets for the exploration of possible cellular functions of ORF proteins through a spectrum of neighboring annotated protein partners. A total of 42 out of 219 presumably cancer-associated ORF proteins and 30 cancer-dependent binary PPIs were found. Additionally, a bibliometric analysis of 204 publications allowed us to retrieve biomedical terms related to ORF genes. In spite of recent progress in functional studies of ORF genes, the current investigations aim at finding out the prognostic value of CxORFx expression patterns in cancers. The results obtained expand the understanding of the possible functions of the poorly annotated CxORFx in the cancer context.

## 1. Introduction

Elucidation of molecular mechanisms underlying the malignant transformation of cells is one of the most popular areas in biomedical investigations with preclinical and clinical implications. Aberrant functioning of signal transduction, regulatory and metabolic cascades significantly contribute to malignant transformation [1], therefore, many ways of pharmacological correction of molecular components of cascades are being under development [2,3,4]. In recent years, significant progress has been made in the identification of proteins, which are directly involved in oncogenic cascades and modulate them by means of protein–protein interactions (PPIs) and post-transcriptional regulation [5,6,7,8]. Thus, the exploration of a spectrum of novel cancer-associated proteins is a promising strategy in anticancer drug discovery.

Currently, up to 90% of predicted human proteins (19,467 proteins) have been detected with high-confidence proteomic methods [9,10] and annotated, while 10% of proteins have not yet been annotated or at least poorly annotated in structural, functional and disease-specific context [10,11]. Among these proteins, there is a group of uncharacterized chromosome-specific open-reading frame proteins (CxORFx or ORF proteins) [12,13,14,15]. The nomenclature of CxORFx genes (ORF genes), encoding ORF proteins, means a chromosome number (Cx) and an open-reading frame number (ORFx). It is worth mentioning that not all mapped ORF genes are protein coding. Of particular interest are studies devoted to the elucidation of the possible involvement of uncharacterized ORF genes in cancers. Some ORF genes such as *C20orf27* [16] and *C19orf53* [17] have already been described in association with cell invasion and proliferation. *C8orf48* gene overexpression has led to a reduction of these biological processes in colorectal cancer cells [18]. C15orf39 protein has been identified as a substrate for mitogen-activated protein kinase MAPK1 [19], which is frequently deregulated in cancers. Recently, Joshi and colleagues [20] showed that C16orf87 protein interacts with epigenetic master regulator histone deacetylase 1 targeted by anticancer drugs. Additionally, there are data on gene expression (e.g., the Human Proteome Atlas) and protein–protein interactions patterns with the participation of ORF proteins. The above-mentioned facts point to a research relevance in functional and disease-specific characterization of ORF genes, which is still missing for the vast majority of them.

A group of 342 ORF genes, belonging to the ‘T-dark’ category, was retrieved from the PHAROS database. A ‘T-dark’ category includes genes (proteins) that were mentioned in publications and other evidence, but their functional roles and pharmacological targeting have not been fully studied or remain unknown. A list of 219 ORF genes that are differentially expressed in a wide range of cancers was selected. Then, we used systems biology and bioinformatic analysis to reveal associations between ORF genes and malignant transformation of cells as well as to predict the involvement of uncharacterized proteins in cellular functions and molecular cascades through a spectrum of neighboring annotated proteins that physically interact and form binary or higher-order complexes with them.

## 2. Results

### 2.1. ORF Genes: Expression Patterns and Their Prognostic Significance 

#### 2.1.1. Gene Expression Patterns

For a panoramic view of 219 differentially expressed ORF genes, we used the principal component analysis, which made it possible to separate at least two cancer clusters (Figure 1) with similar expression patterns of ORF genes. One cluster includes 68 genes predominantly up-regulated in diffuse large B cell lymphoma (DLBC) and thymoma (THYM) (cluster 1). Cluster 2 includes four cancer types: 16 up-regulated genes in uterine corpus endometrial carcinoma (UCEC), 11 up-regulated genes in ovarian carcinoma (OV), 9 genes in colorectal adenocarcinoma (COAD) and 12 genes in rectum adenocarcinoma (READ). More specific expression patterns of ORF genes are characteristic of discrete (non-clustered) cancers. Thus, 124 out of 219 genes are down-regulated in testicular germ cell tumor (TGCT), except for one up-regulated (C6orf132), and 100 out of them are down-regulated in TGCT only (mentioned here, ‘TGCT-specific 100-gene expression signature’, Appendix A). Twenty-three ORF genes are up-regulated in acute myeloid leukemia (LAML) and ten genes are down-regulated in skin cutaneous melanoma (SKCM) (Figure 1). 

There is also a number of ORF genes with different expression in ≥ five cancers (mentioned here, ‘a pan-cancer group’) (Figure 2): preferentially down-regulated genes (*C11orf96*, *C5orf38* and *C8orf88*); preferentially up-regulated genes (*C19orf48*, *C1orf210*, *C4orf48* and *C6orf132*); mixed type of gene expression (*C1orf162*, *C2orf54*, *C2orf74* and *C14orf132*). Figure 2 shows a heat map of gene expression changes at mRNA and total protein levels. It can be seen that there is good consistency between the vectors of expression changes at both levels for *C11orf96*, *C6orf132* and *C8orf88* genes, in particular, in UCEC. 

#### 2.1.2. Prognostic and Predictive Significance 

The tissue-specific transcriptomic signatures are useful for the prognosis of cancer outcomes (survival rates and disease monitoring) as well as for the prediction of the immune and chemotherapy effectiveness. Prioritization of gene expression signatures with prognostic significance will contribute to the development of multigene transcriptomic panels for cancer molecular subtyping and a personalized approach to cancer treatment. Prognostic significance is often connected with the overall survival of patients (OS, the length of time from the date of diagnosis or the start of treatment for disease, during which patients with a diagnosed disease are still alive) and disease-free survival (DFS, the length of time after primary treatment for disease ends that the patient survives without any signs of disease). Kaplan–Meier analysis is a statistical method for estimating the survival curve with a log-rank test, which provides the comparison of groups with low and high levels of gene expression. 

The potential prognostic significance is already known for 85 out of 219 ORF genes encoding chromosome-specific uncharacterized ORF proteins in 16 types of solid tumors (according to the Human Proteome Atlas portal) (Appendix A). 

Endometrial, liver, pancreatic and renal cancers have the largest number of ORF genes with prognostic significance—20, 20, 14 and 43 genes, respectively. In addition, many genes have prognostic significance in more than one type of cancer. For example, a high level of *C3orf62* gene expression correlates with higher survival rates in patients with urothelial/gastric/pancreatic/breast, lung/head and neck cancers, while in the case of liver cancer, it correlates with lower survival rates.

We attempted to find novel transcriptomic signatures of differently expressed ORF genes in cancers using a Kaplan–Meier analysis of The Cancer Genome Atlas cohorts. It was shown that genes included in ‘TGCT-specific 100-gene expression signature’ have no comprehensive prognostic significance in the TGCT cohort, except for a ‘panel’ of six genes—*C11orf94*, *C1orf105*, *C20orf144*, *C22orf31*, *C9orf13* and *CXorf49*. The low levels of their expression correlate with better DFS of patients with TGCT (hazard ratio (HR) = 2.9 and *p*-value = 0.0047 for the high expression group, n = 136). 

Interestingly, in colorectal adenocarcinoma (COAD) or liver hepatocellular carcinoma (LIHC) cohorts, the high levels of gene expression of a total amount of 100 genes correlate with poor OS (HR = 1.7 or 1.8, respectively, *p*-value < 0.05) (Figure 3a,b). Conversely, the low levels of gene expression in pancreatic adenocarcinoma (PAAD) and UCEC correlate with DFS (HR = 0.46–0.48, *p*-value < 0.05) (Figure 3c,d). It should note that with a more stringent cut-off HR-values (0.5 > HR > 2), connections of expression patterns of total amount of 100 genes with DFS are kept in PAAD or UCEC only. Further, we determined that a ‘panel’ of seven genes (*C16orf78*, *C16orf86*, *C16orf96*, *C19orf18*, *C20orf144*, *C22orf42* and *C3orf62*) out of 100 genes make a decisive contribution since their exclusion from analysis leads to a significant decrease in prognostic significance of remaining 93 genes in PAAD or UCEC (*p*-value > 0.05). Moreover, it is found that the low levels of expression of all seven genes, at least in PAAD, correlate with unfavorable DFS (HR = 0.24, *p*-value = 0.0012) and OS (9.8 vs. 16.6 months, HR = 0.37, *p*-value = 0.000018). 

Table 1 shows the potential prognostic significance of several ORF genes that allowed us to prioritize two novel transcriptomic gene signatures for PAAD and READ. First, the high levels of *C14orf119* and *C5orf46* gene expression (gene signature I) correlate with poor OS in PAAD (16.6 vs. 30.4 months, HR = 2.1 (1.37–3.18), *p*-value < 0.001). In addition, this gene signature was specific for LIHC cohort only (14.2 vs. 54.1 months, HR = 2.82 (1.69–4.72), *p*-value < 0.001) (a plot not shown). Second, the low levels of *C6orf132*, *C6orf222* and *C4orf19* gene expression (gene signature II) correlate with poor OS in READ (33.1 vs. 52.7 months, HR = 0.36 (0.16–0.8), *p*-value < 0.01) (a plot not shown). A similar HR value is observed in the kidney renal clear cell carcinoma cohort (KIRC) (HR = 0.44 (0.32–0.61), *p*-value < 0.001) (a plot not shown). Thus, transcriptomic gene signatures I and II can be considered sufficiently specific for these types of cancers with a clear difference in survival times almost twice and higher in low and high-expression groups. From Table 1, it also follows that the expression levels of *C11orf52*, *C9orf116*, *C17orf51* and *C1orf53* genes in the UCEC cohort have prognostic significance, but median five-year survival values were not calculated, probably due to the small number of appropriate clinical cases.

Connections between ORF genes expression levels and immune cell infiltration in the corresponding cancers were also searched for. In general, overexpression of the *C4orf46*, *C9orf40*, *C17orf67* and *C21orf58* gene of thymoma correlates with tumor immune cell infiltration and gene expression levels of *C1orf162*, *C16orf54* and *CXorf21* correlate with immune cell infiltration in many primary and metastatic tumors (Appendix A). A high positive correlation is observed between the up-regulation of some ORF genes and increased tumor immune cell infiltration, especially by dendritic cells. 

Finally, connections between ORF gene expression in tumor tissues and responses to chemotherapy (predictive significance) were investigated. *C22orf42* gene expression levels in OV correlate with good responses to platinum-based drugs (fold-change and AUC values ≈ 2 and 0.7, respectively) (Appendix A).

#### 2.1.3. Other Aspects of Differently Expressed ORF Genes

We prioritized a group of differentially expressed ORF genes in cancers, for which the occurrence of significant effects on the viability of a limited number of cell lines (“strongly selective genes” subgroup) or most cell lines (“common essential genes” subgroup) is directly dependent on gene knockouts or knockdowns (Table 2). Up-regulation of some ORF genes in cancers is consistent with activity changes of certain protein kinases that are encoded by driver genes and involved in signaling pathways responsible for cancer progression. The cancer-dependent regulation of *C12orf49* gene expression through nuclear transcription factor Y subunit alpha (NFYA) is not yet known, however, there is evidence of its role in the trans-activation of some cancer-promoting genes [21,22]. Transcription factor GATA3, which can bind to the *C8orf76* gene promoter, is expressed at an early stage of thymus development and participate in thymocyte differentiation [23] as well as in control lymphoid cell differentiation, being a tumor suppressor in B-cell lymphomagenesis. It also follows from Table 2 that at least *C1orf109* and *C8orf33* gene expression may depend on binding in their promotor regions of cAMP-responsive element binding protein 1 (CREB1). Since CREB1 functions in cancer signaling [24], together with data on the involvement of ABL1 and ERBB2 protein kinases in pathways in cancers, it suggests the presence of upstream cancer-associated tissue-specific transcriptional regulation in DLBC and THYM cancers. Thus, an upstream regulation of the expression of ORF genes, which are critical for cell viability, can occur during malignant transformation of cells. 

### 2.2. ORF Proteins 

Evidence on a protein level is found for 200 out of 219 ORF genes according to immunohistochemistry and/or mass-spectrometry data. Nineteen genes (*C1orf14*, *C1orf53*, *C1orf115*, *C2orf81*, *C4orf45*, *C5orf66*, *C6orf48*, *C6orf99*, *C7orf71*, *C8orf49*, *C10orf25*, *C11orf94*, *C11orf98*, *C16orf82*, *C16orf90*, *C16orf197*, *C19orf67*, *C20orf141* and *C20orf173*) are expressed at a transcript level only, pointing to probable ‘missing’ proteins. For example, the functionally significant product of the *C8orf49* gene is a long non-coding RNA [25,26]. The distribution of ORF proteins by molecular weights (Mw) is shown in Appendix A. The major part (25%) includes proteins with Mw from 20 to 30 kDa. In total, micro-proteins with Mw from five to 20 kDa are almost 38%. It is known that micro-proteins possess a number of beneficial biological activities, for example, C5orf46 protein possesses antimicrobial activity [27]. It is also interesting that the micro-protein C19orf48 is widely expressed in cancers and can be processed for cytotoxic T lymphocyte (CTL) recognition [28].

Data on 237 subcellular localization sites were retrieved from The Human Protein Atlas for 135 out of 200 ORF proteins. Two major groups of proteins with nuclear localization sites (nucleoplasm, nucleolus, nuclear membrane and nuclear bodies) and cytoplasmic sites make up about 41% and 18%, respectively (Appendix A). It can also be noted that seven (C1orf54, C4orf48, C5orf46, C17orf67, C17orf99, C21orf62 and C22orf15) out of 135 proteins are predicted to be secreted. Using the Exocarta server [29,30], seven other secreted ORF proteins are additionally identified: exosome-derived proteins C2orf16, C2orf74, C11orf52, C16orf87 and C19orf18 (from urine samples); C1orf198 and C2orf88 proteins (from blood samples). C2orf74 and C19orf18 proteins appear to be membrane associated since they are predicted to contain typical alpha helices for anchoring in the hydrophobic membrane bilayer. Thus, at least 14 secreted ORF proteins may be involved in intercellular communication and signaling functions. We did not observe data on these secreted ORF proteins as potential biomarkers.

### 2.3. Protein–Protein Interactions of ORF Proteins and Their Subinteractomes

#### 2.3.1. Structural Data

There is still very little information on the protein 3D structures among 219 differentially expressed ORF genes in cancers. The Protein Data Bank, URL: https://www.rcsb.org/ (accessed on 15 February 2023) contains two crystallographic 3D-models of C1orf123 monomeric protein (PDB ID: 5zlq) [31] and C9orf64 dimeric protein (PDB ID: 7ugk) [32]. Using the interactive tool PDBePISA v.1.52 for analysis of protein–protein interfaces, URL: https://www.ebi.ac.uk/msd-srv/prot_int/cgi-bin/piserver (accessed on 17 February 2023), we analyzed the last model with resolution of 1.78 Å. Despite the relatively small interface area (6% of the total surface area), it is enriched with 13 hydrogen bonds and 10 salt bridges. Interestingly, the dimeric form of C9orf64 protein, according to CavityPlus server [33]) has two predicted ligand-binding pockets on the surface with a high level of drugability (drug scores are 884 and 2237) compared to monomeric form with one ligand-binding pocket (drug score = 281). This implies the higher potential of C9orf64 protein to pharmacological targeting its dimeric form than monomeric one in case of its valid clinical significance. Protein C2orf76 is described as a dimeric protein [34] that is involved in the formation of an aggressive pancreatic cancer phenotype through induction of cell invasion and proliferation. Protein C2orf76 can be considered as a potential drug target, but there is no crystallographic data for in silico drug design. 

#### 2.3.2. Subinteractomes of ORF Proteins

The term ‘subinteractome’ means here a set of all the known protein partners that physically interact with a target protein. What is known about the intermolecular interactions with the participation of target ORF proteins? Protein C1orf123 specifically interacts with the heavy-metal-associated domain of a copper chaperone for superoxide dismutase encoded by the CCS gene [31]. Participation of another protein C16orf62 in the formation of a functionally significant heterotrimer with DSCR3 and VPS29 proteins is shown in [35]. Protein C11orf98 can be a putative interaction partner of nucleophosmin (NPM1) and nucleolin (NCL) [36]. Analysis of structural features and the PPIs spectrum of the C11orf96 protein points to the fact that it may be phosphorylated and plays a role in endoplasmic reticulum stress, protein ubiquitination and gene transcription [37]. Protein C1orf112 is probably cancer-associated as it follows from the functional analysis of its 31 protein partners and co-expression data [38]. Thus, uncharacterized ORF proteins can participate in homo and heterodimeric protein interactions and deciphering their subinteractomes helps to predict functional roles through analysis of the PPIs spectrum. 

A number of interactomic browsers (BioPlex, HIPPIE, MINT, hu.MAP, IID, InnateDB and HuRI), as well as interactomic datasets from the publications (see Section 4 in more details), were used to collect information about the subinteractomes of each ORF protein. In general, 4317 unique PPIs for 177 ORF proteins were found, of which eight PPIs are between the ORF proteins themselves. These are eight binary interactions: C1orf94 and C1orf109; C1orf112 and C11orf87; C1orf52 and CXorf56; C2orf74 and C20orf27; C5orf24 and C16orf71; C5orf51 and C10orf82; C1orf131 and C16orf87; C1orf131 and C20orf197. As for the rest of the 42 ORF proteins, there is a lack of any information about their PPIs. There are 517 protein partners forming binary interactions with two different ORF proteins (potential heterotrimeric complexes) and 768 PPIs, in which one protein partner interacts with three or more ORF proteins (potential multimeric complexes). It indicates a wide PPI subnetwork with connections between ORF proteins through common protein partners and the shortest paths. The last ones are among potential multimeric complexes, where ORF proteins can indirectly interact with the first-, second- and higher-order protein partners (Figure 4). 

As a rule, subinteractome of a target protein includes protein partners that form mainly stable PPIs with it, which is due to the peculiarities of experimental approaches used for isolation and identification of protein partners from biosamples. At the same time, it is not completely known what proportion of protein partners form direct interactions with a target protein. Therefore, analysis of subinteractomes was carried out on the assumption that each protein partner has an equal probability to directly interact with a target ORF protein. The distribution of a number of protein partners of ORF proteins within 177 subinteractomes are as follows: <7 protein partners (63 ORF proteins), from 7 to 15 (30 ORF proteins), from 15 to 30 (40 ORF proteins), from 30 to 70 (27 ORF proteins), from 70 to 100 (6 ORF proteins) and over 100 protein partners (11 ORF proteins). So, subinteractomes of 103 ORF proteins, each containing from 7 to 100 protein partners, are chosen for analysis. The choice of the minimal number of protein partners is due to the sufficiency of functional enrichment analysis. The maximal number is limited in order to reduce the redundancy of output functional terms. 

Protein partners included in the subinteractome of any ORF protein are analyzed for direct or indirect associations with ‘cancer hallmarks’. The ‘cancer hallmark’ score (CH-score), which means here the relative frequency of occurrence of CH-associated proteins per 10 protein partners in the subinteractome of an ORF protein, is calculated. For example, CH-score equal to one means that only one CH-associated protein is present among 10 protein partners, including itself. Next, 70 out of 103 subinteractomes are prioritized according to CH-score value ≥ 2, which corresponds to approximately the 25th percentile at a median CH-score value equal to 2.8. Such subinteractomes consist of CH-associated proteins (a ‘core’) and other non-CH-associated protein partners. The results of molecular pathway enrichment analysis of ‘core’ proteins and PPIs cluster modeling as well as tissue-specific co-expression analysis are shown in Appendix A. At least 42 subinteractomes of ORF proteins are enriched with molecular pathways in cancers, form highly connected PPIs clusters of CH-associated proteins and have noticeable or high correlation of co-expression between ORF genes and genes, encoding CH-associated proteins (Appendix A). These facts indirectly speak in favor of the involvement of ORF proteins in signaling cascades and regulatory networks, which are critical for the malignant transformation of cells.

High correlation coefficients (r-values ≥ 0.85) of gene co-expression may additionally hint at the presence of direct interactions between ORF proteins and their protein partners. As it follows from Appendix A, among such tissue- and cancer-specific PPIs, there are the following hits: C2orf88 and PRKAR1A (brain tissue), C5orf24 and WDR45B (thymoma), C5orf24 and GOPC (thymoma), C6orf132 and CDH1 (colon transverse), C6orf132 and EZR (colon), CXorf56 and RPRD1B (thymoma) and C16orf87 and ANAPC10 (thymoma). In this regard, it is interesting to find potential binary PPIs, whose gene co-expression patterns are highly cancer dependent. These PPIs can be predicted by the change of correlation coefficients from positive to negative values in cancers relative to normal tissue and vice versa. Thus, a group of 30 potential tissue-specific binary PPIs, involving ORF proteins and CH-associated protein partners, is shown in Table 3. It follows that 23% of PPIs belong to the subgroup of cancer-dependent PPIs in cancers (positive r-values of gene co-expression). This subgroup is represented with PPIs between ORF proteins and known oncoproteins, for example, C1orf131/JUN (transcription factors), C5orf24/PBX1 (pre-B-cell leukemia transcription factor 1), C2orf88/PRKACA (cAMP-dependent protein kinase catalytic subunit alpha), C6orf132/CDH1 (fizzy-related protein homolog) and C6orf132/EZR (tumor suppressor ezrin). Ezrin, being an adapter protein between the actin cytoskeleton and the plasma membrane, is involved in cancer promotion through the modulation of signaling pathways [39,40]. However, data on the involvement of ezrin in TGCT cancer is absent. Thus, we demonstrated that the interaction of a number of ORF proteins with oncoproteins are cancer-dependent PPIs.

Another subgroup of PPIs presumably perturbed by cancers is characterized by negative or near-zero r-values in cancers and positive r-values of gene co-expression in conditionally normal tissues, for example, C1orf131/EIF3L, C19orf53/APEX1, C1orf198/PPP2R1A and C1orf198/ARHGEF1 (Table 3). It should be noted that a significant part of protein partners of ORF proteins in this subgroup are protein-kinases and E3-ubiquitin-protein ligases that function as universal modulators of the posttranslational activity of protein substrates and modulators of the cellular GTPase system. Impaired interaction of C19orf44 protein with the testis-specific multifunctional enzyme GAPDHS (glyceraldehyde-3-phosphate dehydrogenase, 44 PPIs in the BioGRID database, ≥2 evidence) may be associated with modulation of spermatogenesis and male fertility [41] or metabolic reprogramming in cancers via GAPDH [42]. Thus, using the gene co-expression analysis, we demonstrate that a number of ORF proteins may participate in cancer-dependent PPIs or PPIs perturbed by cancers.

## 3. Discussion

Elucidation of cellular roles of uncharacterized chromosome-specific ORF proteins is the focus of many scientific investigations. We found at least 27 publications (see Table 4) devoted to the functional annotation of ORF genes mainly through gene knockdowns or knockouts and other experimental approaches. In Table 3, it follows that there is a large biodiversity of established or proposed cellular roles of 20 ORF proteins: involvement in signaling pathways and processes such as ciliary motility and ciliogenesis, protein folding and degradation, lipid homeostasis, cell cycle regulation, protein trafficking, mitophagy and apoptosis (Table 4). To explore the publication landscape on 219 target ORF genes or ORF proteins in the cancer context, a bibliometric analysis was performed using the VosViewer software v.1.6.18. A full-text search allows us to select 204 publications, in which 92 out of 219 differently expressed ORF genes are mentioned in association with the malignant transformation of cells. Thus, *C9orf50*, *C9orf64*, *C5orf66*, *C16orf74* and *C10orf55* genes are most mentioned in the cancer context (each gene was mentioned at least in six publications), while 42 out of 92 genes are mentioned in one publication only. A co-occurrence map of 33 terms retrieved from the abstracts and titles of 204 found publications is shown in Figure 5, where these terms are divided into six thematic clusters, containing such terms as ‘patient survival’, ‘staging’, ‘progression’, ‘recurrence’ and ‘predictive significance’. In most publications, ORF genes are described mainly as a part of transcriptomic or methylation signatures correlating with disease prognosis. In a much smaller number of publications, specific cancer-associated features of some ORF genes were studied, for example, the capability to form gene fusions [43], participation in the PPI subnetwork as hub genes [44] and elucidation of upstream regulators (e.g., oncogene micro-RNA miR-556 for *C8orf48* gene [18]). Thus, the bibliometric analysis makes it possible to observe any associations with cancers for half of the target ORF genes mentioned in at least one publication. The conclusion is that the current vector of investigations of uncharacterized chromosome-specific ORF genes is to identify their prognostic significance.

At the same time, cellular roles remain unknown for many ORF proteins under both physiological and pathological conditions, so one of the ways to predict them is to analyze the already-known functions of protein partners that physically interact with uncharacterized ORF proteins. Functional terms, describing subinteractomes of ORF proteins, are shown in Appendix A. For example, the protein partners of secreted C2orf88 protein are related to post-translational modifications mediated through protein kinase A as well as important cellular functions such as ciliary motility. C4orf17 protein’s function may be associated with chromatin modifications through histone methylation, which follows from the high representation of ATRX, CHD3, KAT2B, KDM1A, PRMT1 and SUV39H1 proteins in the C4orf17’s subinteractome. Protein CXorf56 along with its interacting partner C1orf52 may be involved in the formation of spliceosome complex. The vast majority of interacting partners of C16orf90 protein are responsible for post-translational modifications (neddylation-dependent protein degradation (BIRC2, NDFIP1, NDFIP2 and XIAP) and ubiquitin-dependent protein degradation (BIRC2, RNF123, TRAF2 and XIAP). In this regard, the C16orf90 protein may act as a part of protein-modifying enzymes complex. Protein C11orf98 is involved in the regulation of DNA-binding transcription factor activity that stems from the spectrum of its protein partners (ESR1, ESR2, FOXA1, JUN and WWP2). On the other hand, functional coupling of subinteractomes of different ORF proteins may indicate identical GO terms (Appendix A): ‘GO:0000151~ubiquitin ligase complex’ (C16orf87 and C6orf222), ‘GO:0016197~endosomal transport’ (C11orf49 and C1orf210), ‘GO:0016311~dephosphorylation’ (C1orf21, C2orf74 and C20orf27), ‘GO:0016607~nuclear speck’ (C1orf226 and C19orf53), ‘WP254~apoptosis’ (C10orf67 and C16orf90), ‘PA443358~aneuploidy’ (C1orf226, C6orf222 and C16orf87). The most common term ‘GO:1901987~regulation of cell cycle phase transition’ is typical for subinteractomes of C16orf87, C3orf62, C16orf87, CXorf56, C11orf49 and C16orf87 proteins. 

It is important to find out whether ORF proteins are associated with signaling pathways in normal and pathological conditions. The separate over-representation analysis is performed for each subinteractome of ORF protein. It follows from Appendix A that there are a variety of functional terms covering ≥25% of the total amount of protein partners in the subinteractome. The subinteractome of C1orf21 protein is enriched with the ‘WP3932~focal adhesion-PI3K-Akt-mTOR-signaling pathway’ term. This pathway connects the membrane receptor and actin cytoskeleton through a cascade of protein kinases and phosphatases and affects the cell shape and motility. In neoplastic cells, aberrant activation of this pathway leads to the formation of a drug-resistant cancer phenotype [70,71]. An analysis of 79 candidate protein partners of C11orf52 points to its involvement in the Wnt-signaling pathway through desmoglein 1 (DSG1) [72]. In this regard, the subinteractome of C11orf52 protein contains HRAS (‘Harvey RAt Sarcoma virus oncogene’), which is one of the most mutated genes in solid tumors and SRSF protein kinase 1 (SRPK1), which was identified as part of HRAS signaling [73]. Thus, both C1orf21 and C11orf52 proteins can theoretically act as adapter proteins or modulators of these signaling pathways.

There is an opportunity for ORF proteins to interact with each other. Assumptions on binary interactions between C2orf74 and C20orf27, C5orf24 and C16orf71 proteins are strengthened by some interesting findings. First, the subinteractomes of these ORF proteins contain proteins associated with ‘cancer hallmark’ and oncoproteins (e.g., the case of C5orf24 and its protein partners XPO1, PBX1, MYC, CIC, GOPC, CREB1 and STK11, Appendix A). Second, a comparative analysis of subinteractomes indicates the existence of 29 and 14 common protein partners for potential interactions C2orf74/C20orf27 and C5orf24/C16orf71, respectively. Over-representation analysis shows the related GO terms for both ORF proteins in each pair: ‘GO:0016311~dephosphorylation’ and ‘GO:1903293~phosphatase complex’ (C2orf74 and C20orf27) as well as ‘PA444750~leukemia’ and ‘PA444761~ leukemia, myeloid’ (C5orf24 and C16orf71) (Appendix A). As for another potential interaction C1orf131/C16orf87, only histone acetyltransferase KAT5 [74] is a common protein partner that can post-translationally modify both ORF proteins. The subinteractome of C16orf87 protein contains HDAC1 and HDAC2 histone deacetylases as well as 16 different histones, thus indicating functional associations of C16orf87 protein with epigenetic modulation of chromatin. *C1orf131* gene belongs to a subgroup of common essential genes (Table 2) and its subinteractome is over-represented by the term ‘GO:0022613~ribonucleoprotein complex biogenesis’ covering 11 out of 43 protein partners (25%) (Appendix A). 

Another point is that genetic alterations (GAs) more frequently occur in cancer-associated genes. Analysis of the cBioPortal pan-cancer cohort shows that the median GA frequency for a sample of 159 out of 219 differentially expressed ORF genes is about 1.3%. Gene amplifications are the prevalent type of GAs. It is interesting to note that GA frequency is up to 6% in *C3orf70*, *C8orf33*, *C8orf76* and *C8orf82* genes according to the most affected cancer types. By the way, an average proportion of GAs for known oncogenes (e.g., *KIT*, *MYC*, *CTNNB1*, *MDM2* and *APC*) in the same pan-cancer cohort ranges from four to 9%. At least the function of genetically unstable *C8orf76* is associated with cell proliferation [55], so amplification of the *C8orf76* gene in cancers may be an independent risk factor of enhanced cell proliferation. According to the Cancer Gene Census portal, only the *C15orf65* gene contains driver somatic mutations in primary mediastinal B-cell and Hodgkin’s lymphoma. Currently, the ClinVar database, URL: https://www.ncbi.nlm.nih.gov/clinvar/ (accessed on 23 March 2023), aggregates records of missense mutations in these ORF genes with ‘uncertain’ or ‘likely benign’ clinical significance. A search for novel genetic variants of *C3orf70*, *C8orf33*, *C8orf76* and *C8orf82* associated with systemic diseases, including cancers, is of practical interest for medical genomics. 

Identified gene expression signatures involving chromosome-specific ORF genes with potential prognostic and/or predictive significance can be further translated into multigene transcriptomic panels to predict disease dynamics or tumor responses to therapeutics. There is plenty of patents disclosing such panels, e.g., WO2021164492A1 (*C10orf99*), WO2018097166A1 (*C17orf67*) and WO2020226333A1 (*C14orf45*, *C16orf61*, *C7orf63*, *C10orf76* and *C12orf72*) in the Espacenet database (https://worldwide.espacenet.com). In this study, two highly specific gene expression signatures (*C14orf119* and *C5orf46*) and (*C16orf78*, *C16orf86*, *C16orf96*, *C19orf18*, *C20orf144*, *C22orf42* and *C3orf62*) were found for PAAD that are associated with the overall survival of patients and disease-free survival, respectively. Therefore, verification of gene expression signatures in a larger cohort of patients (balanced by subgroups with different cancer stages, grades, mutation burden, etc.) will provide the design of a clinically relevant transcriptomic panel for molecular subtyping of one of the highly aggressive solid tumors. 

We focused on fourteen secreted ORF proteins. Peptide compositions of, at least, seven proteins (C4orf48, C5orf46, C22orf15, C2orf16, C11orf52, C1orf198 and C2orf88) were identified by mass spectrometry in human blood plasma or urine samples according to the Peptide Atlas portal, URL: https://db.systemsbiology.net/sbeams/cgi/PeptideAtlas/ (accessed on 21 May 2023). So far, the diagnostic significance of these proteins in the ‘liquid biopsy’ paradigm for cancer detection has not been studied, which also follows from the materials of the collaborative program Early Detection Research Network (EDRN) [75], and the above mentioned secreted ORF proteins may represent novel molecular entities in this scientific field.

Among other biological features of some uncharacterized chromosome-specific ORF proteins is the high representation of oncoproteins (encoded by highly mutated genes in cancers) in their subinteractomes. This fact may indirectly indicate the involvement of ORF proteins in cancer-associated signaling pathways in yet unknown roles. These include eight ORF proteins: CXorf56 (BRD4, EZH2, JUN, LMNA, MYC, NSD2, RECQL4, RET); C8orf48 (CCNE1, FGFR3, HRAS and TSC1); C6orf132 (CDH1, ESR1 and EZR), C5orf24 (CIC, CREB1, GOPC, MYC, PBX1, STK11 and XPO1); C1orf226 (BARD1, BCR, BRD3, FOXO3, HRAS and PRCC); C16orf71 (CDKN2C, CREB1, EZH2, PBX1, PTEN); C11orf98 (FOXA1, JUN and MYC); C1orf123 (CDKN1A, CLTC, HEY1 and RAF1); C1orf131 (CDC73, JUN, KRAS, PPP2R1A and SMARCB1). Thus, functional annotation of these ORF proteins via the expansion of data on the tissue-specific and cancer-specific spectrum of protein–protein interactions will clarify their roles in signaling pathways and, possibly, allow us to select new molecular targets for pharmacological interventions.

## 4. Materials and Methods

### 4.1. Gene Expression Data Analysis

PHAROS database v.3.15, a web-based interface for exploring and visualizing the Target Central Resource Database (TCRD) data generated by the Illuminating the Druggable Genome project (IDG) [76], was used for retrieval of ORF gene list. 

Web-based server Gene Expression Profiling Interactive Analysis (GEPIA2) [77] adapted for the analysis of The Cancer Genome Atlas (TCGA) cohorts was used for selection of differentially expressed genes (DE-genes) in cancer and normal tissues according to the following settings: |log2FC = 2| (four-fold change), *p*-value cut-off < 0.05, match TCGA normal and GTEx (Gene Tissue Expression) datasets. 

Survival analysis (Kaplan-Meier plots) was carried out using the web-server KMplotter [78]. 

ROCplotter server [79] was used to explore the gene expression patterns and response to therapy of patients with glioblastoma, breast, ovarian and colorectal cancers. AUC (Area Under the Curve) and gene expression fold change values equal to 0.7 and 2.0 were set as cut-off values, respectively.

TIMER (Tumor IMmune Estimation Resource) server is a comprehensive resource for systematical analysis of immune infiltrates across diverse cancer types [80,81]. “Gene” or “Survival” modules were used to explore the correlation between gene expression patterns and abundance of immune infiltrates as well as between immune infiltrates and clinical outcomes.

Patterns of genetic alterations (missense and truncating mutations, and copy-number variations (deletion or amplification)) were explored in pan-cancer MSK-IMPACT Clinical Sequencing Cohort (10,953 patients/10,967 samples) cBioPortal server [82].

A Cancer Dependency Map or DepMap portal, URL: https://depmap.org/portal/ (accessed on 1–5 February 2023), allows for defining genes that are essential for cell viability using data on systematic loss-of-function screens (RNA interference and CRISPR-Cas9) in the cell lines representing the heterogeneity of tumors [83]. We used the DepMap portal for definition of ORF genes as “common essential genes” or “strongly selective genes”, implying the significant effects of gene depletion studies in most or several of the screened cell lines, respectively. 

Gene expression changes following kinase perturbations assays (inhibition, activation, knockdown, knockout, over-expression and mutation) from the GEO datasets (Gene Expression Omnibus) were explored using kinase perturbations data annotated by the Ma’ayan lab [84], which are available at EnrichR database, URL: https://maayanlab.cloud/Enrichr/ (accessed on 23–25 February 2023).

Data on the interactions of transcription factors (TFs) with the promotor regions of ORF genes was retrieved from EnrichR database (section ‘Transcription’, subsections ‘ENCODE and ChEA Consensus TFs from ChIP-X’, ‘TRANSFAC and JASPAR PWMs’ and ‘ENCODE TF ChIP-seq 2015’). The resulting list contained matched TFs in all three subsections.

### 4.2. Protein Expression Data analysis

The Human Proteome Atlas (HPA) [85] was used for exploration of protein subcellular localization. In case of absence of available data, predictions were made in the Compartment web-server [86], URL: https://compartments.jensenlab.org/ (accessed on 29–31 January 2023), which also integrates the results of automat text mining.

Data on protein expression in normal tissues were obtained from the PepPSy server [87], URL: http://peppsy.genouest.org/query (accessed on 6 February 2023). This server enables the determining in which human tissues investigators should look for unseen (‘missing’) proteins. 

Data on the relative protein abundance in cancer and normal tissues were retrieved from the National Cancer Institute’s Clinical Proteomic Tumor Analysis Consortium (CPTAC) and the National Cancer Institute’s International Cancer Proteogenome Consortium (ICPC) datasets using cProSite web-based tool [88], URL: https://cprosite.ccr.cancer.gov/ (accessed on 20 January 2023). 

The TOPCONS web-server was used to predict transmembrane topology and membrane domains (TM-helixes) of proteins [89], URL: https://topcons.cbr.su.se/ (accessed on 3 February 2023).

### 4.3. Protein Subinteractome Analysis

Data on human PPIs (evidence type—physical interactions) were retrieved from several interactomic browsers: BioPlex v.3.0 [90], Human Integrated Protein–Protein Interaction Reference or HIPPIE v.2.3 (cut-off score value > 0.7) [91], The Molecular INTeraction database (MINT) [92], Human Protein Complex Map (hu.MAP 2.0) [93], ‘NURSA Human Endogenous Complexome’ and ‘PPI Hub Proteins’ in the ‘Pathways’ of Enrichr database [94,95], Integrated Interactions Database (IID v.2021-05) [96], InnateDB [97], reference interactome map of human binary protein interactions (HuRI) [98] and interactomics profiling data from publications [99,100].

Over-representation analysis (ORA) of a list of gene names was performed using the WebGestalt server (WEB-based Gene SeT AnaLysis Toolkit) [101] with gene ontology terms (biological process noRedundant, molecular function noRedundant and cellular component noRedundant), disease terms (DisGeNet and GLAD4U databases), phenotype terms (Human Phenotype Ontology database), network terms (CORUM database) and pathway terms (Wikipathway cancer database) with the following settings: reference gene list—‘genome protein-coding’; minimum number of genes for a category—‘4’; multiple test adjustment—‘Benjamini-Hochberg’; significance level—‘FDR < 0.1’; redundancy reduction—‘affinity propagation’. A term, covering >10% of the total number of proteins in the subinteractome of an ORF protein, was considered a significant term. 

CancerGeneNet [102], URL: https://signor.uniroma2.it/CancerGeneNet/ (accessed on 1–20 March 2023), is a web-server aiming at linking frequently mutated genes in cancers with cancer phenotypes and allows searching for a graph path between any gene-of-interest and ‘hallmarks of cancer’ (i.e., sustaining proliferative signaling, evading growth suppressors, resisting cell death, enabling replicative immortality, inducing angiogenesis, activating invasion and metastasis) [103,104]. CancerGeneNet was used to select protein partners of each ORF protein by associations with ‘hallmarks of cancer’.

PathwAX II, a web-server for pathway annotation based on crosstalk derived through FunCoup’s genome wide functional association networks [105], URL: https://pathwax.sbc.su.se/ (accessed on 15–28 February 2023), was used to explore the pathways that involve proteins, selected by CancerGeneNet web-server, with the following settings: database source—‘KEGG v.94.1’; filter—‘enrichment’, ‘depletion’; multiple hypothesis test correction—‘Benjamini-Hochberg’ and cut-off = 0.01.

The Cancer Gene Census (CGC) database aims to catalogue genes containing mutations that have been causally implicated in cancers (i.e., the driver genes) [106]. CGC was used to search for driver genes, encoding protein partners of ORF proteins, selected by the CancerGeneNet web-server.

Selection of gene co-expression hypotheses in normal and cancer tissues was performed using the GEPIA2 web-server [77] with the following settings: Spearman correlation coefficient > |0.5| at *p*-value < 0.05.

The FunCoup—Functional Coupling framework integrates 10 different evidence types derived from high-throughput genomics and proteomics data in a naive Bayesian integration procedure [107]. The FunCoup v.5.0, URL: https://funcoup5.scilifelab.se/ (accessed on 21–28 March 2023), was used for PPIs subnetwork construction with the following settings: species—‘Homo sapiens’, confidence score—‘0.7’, expansion depth—‘0’, nodes per expansion step—‘30’, evidence—‘Homo sapiens’, link types—‘all types’. 

### 4.4. Others

The search for full-text publications indexed in the PubMed Central system was performed using the Litsence text-mining tool [108]. 

Bibliometric analysis and construction of co-occurrence maps of terms in research type publications were performed using the VOSviewer software v.1.6.18 [109]. 

Principal component analysis was performed using ClustVis [110].

## 5. Conclusions

Currently, there is no detailed understanding of the molecular mechanisms underlying many systemic diseases, as well as malignant transformation of cells. Chromosome-specific open reading frames genes, which functions are not yet fully characterized, may be involved in carcinogenesis. In this work, we performed a systems biology analysis of 219 target ORF genes that are differentially expressed in various cancers. ORF gene expression patterns correlate with disease prognosis, tumor infiltration by immune cells and responses to chemotherapy. Analysis of individual subinteractomes of ORF proteins revealed interactions with a number of proteins, which are associated with ‘hallmarks of cancer’ as well as proteins encoded by highly mutated genes and mapped in cancer-associated signaling cascades. Some of the ORF proteins are involved in cancer-dependent PPIs and PPIs perturbed by cancers and may have functions related to their protein partners. The findings represent an analytical ‘cross-section’ of the current background on uncharacterized ORF genes in the cancer context that is important for prioritizing further fundamental research in this field.

## Figures and Tables

**Figure 1 ijms-24-10190-f001:**
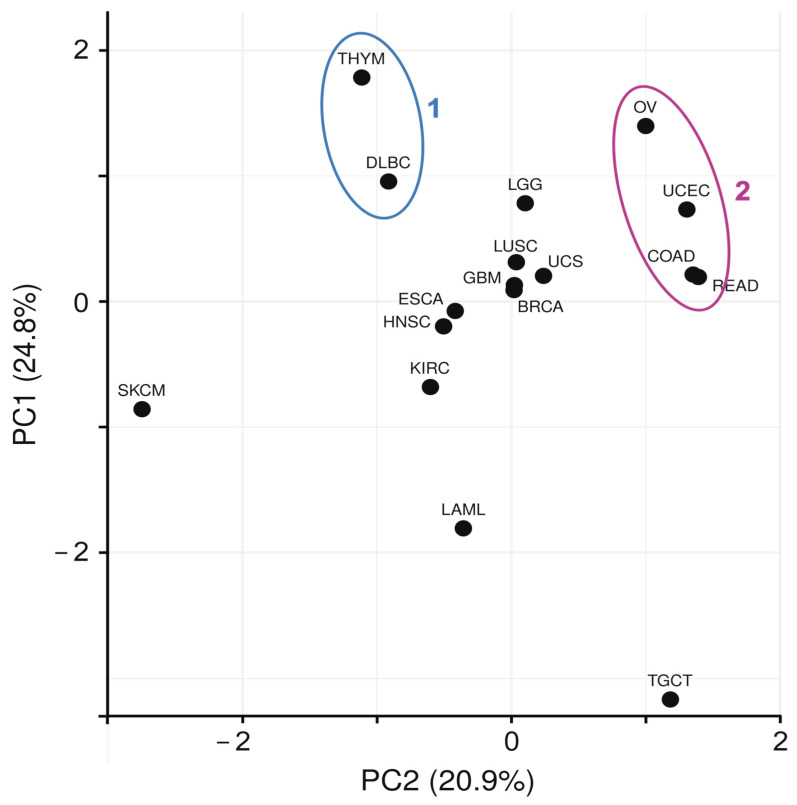
A panoramic view of clustered and non-clustered cancers according to the patterns of differentially expressed ORF genes in the coordinates of principle components PC2 and PC1. No scaling is applied to rows; singular value decomposition (SVD) with imputation was used to calculate PCs. *X*- and *Y*-axis show PC2 and PC1 that explain 20.9% and 24.8% of the total variance, respectively. Clusters 1 and 2 are highlighted in blue and purple, respectively.

**Figure 2 ijms-24-10190-f002:**
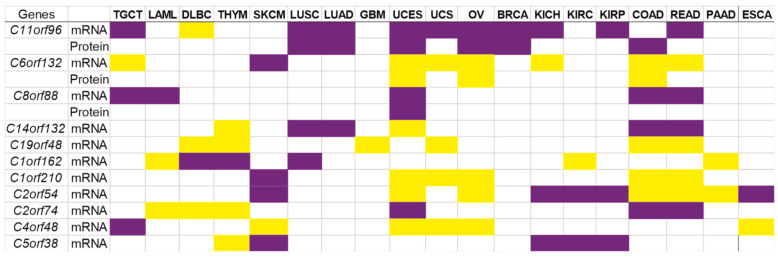
Visualization of differently expressed ORF genes in cancers (pan-cancer group). Up- and down-regulated genes are highlighted with yellow and violet colors, respectively. TGCT—testicular germ cell tumor, LAML—acute myeloid leukemia, DLBC—lymphoid neoplasm diffuse large B-cell lymphoma, THYM—thymoma, SKCM—skin cutaneous melanoma, LUSC—lung squamous cell carcinoma, LUAD—lung adenocarcinoma, GBM—glioblastoma multiforme, UCES—uterine corpus endometrial carcinoma, UCS—uterine carcinosarcoma, OV—ovarian serous cystadenocarcinoma, BRCA—breast invasive carcinoma, KICH—kidney chromophobe carcinoma, KIRC—kidney renal clear cell carcinoma, KIRP—kidney renal papillary cell carcinoma, COAD—colon adenocarcinoma, READ—rectum adenocarcinoma, PAAD—pancreatic adenocarcinoma, and ESCA—esophageal carcinoma.

**Figure 3 ijms-24-10190-f003:**
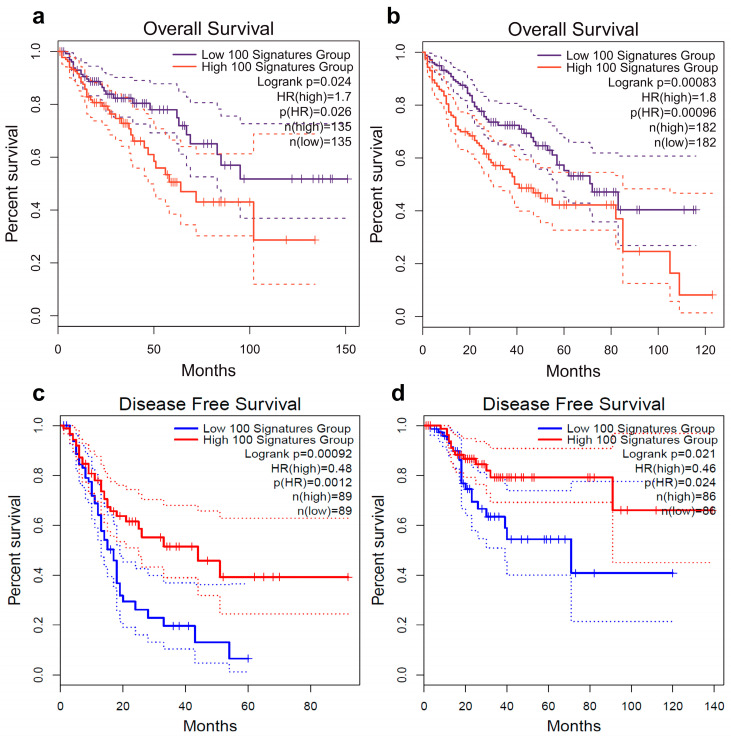
Kaplan–Meier analysis of ‘TGCT-specific 100-gene signature’: (**a**)—colorectal cancer TCGA cohort (OS, n = 135 cases), (**b**)—liver hepatocellular carcinoma TCGA cohort (OS, n = 182 cases), (**c**)—pancreatic adenocarcinoma TCGA cohort (DFS, n = 89) and (**d**)—uterine corpus endometrial carcinoma TCGA cohort (DFS, n = 86). Original data output returned from GEPIA2 web-server. TCGA—The Cancer Genome Atlas.

**Figure 4 ijms-24-10190-f004:**
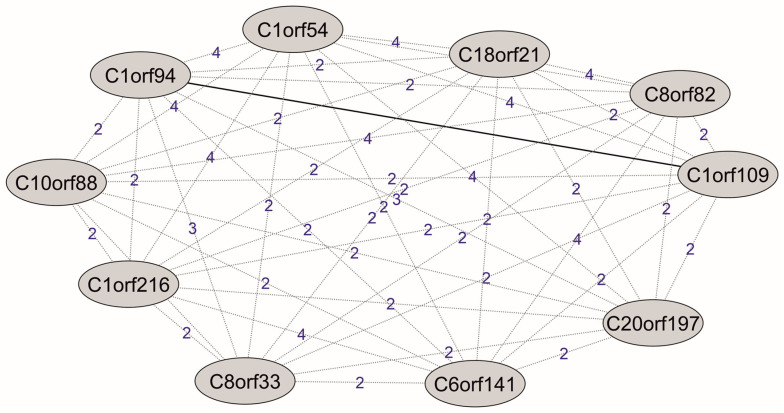
Visualization of the shortest paths between ORF proteins in the PPI subnetwork. The solid line indicates the direct interaction between ORF proteins. The dotted lines indicate the paths between ORF proteins through a number of other interacting proteins in the subnetwork.

**Figure 5 ijms-24-10190-f005:**
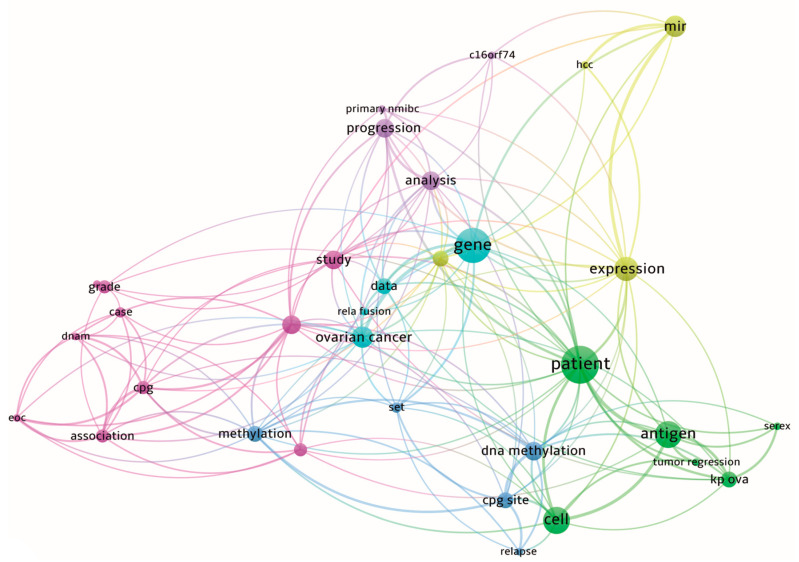
A co-occurrence map of 33 terms retrieved from abstracts and titles of publications. Clusters of terms are highlighted with different colors.

**Table 1 ijms-24-10190-t001:** Prognostic significance of ORF genes expression patterns in different cancers * (five-year overall survival rates).

Gene	Cancer	Hazard Ratio	Low Expression Group, Months **	High Expression Group, Months **	Log-Rank *p*-Value
*C14orf119*	PAAD	1.76 (1.16–2.68)	24.4	17.0	0.0069
*C5orf46*	PAAD	2.1 (1.27–3.46)	16.2	9.77	0.0031
*C6orf222*	READ	0.41 (0.19–0.89)	24.5	47.8	0.0210
*C4orf19*	READ	0.44 (0.21–0.96)	33.1	52.2	0.0350
*C6orf132*	READ	0.37 (0.17–0.8)	33.1	52.2	0.0083
*C11orf52*	UCEC	0.42 (0.24–0.75)	n/d	n/d	0.0023
*C9orf116*	UCEC	0.28 (0.14–0.58)	n/d	n/d	0.0003
*C17orf51*	UCEC	2.51 (1.49–4.34)	n/d	n/d	0.0006
*C1orf53*	UCEC	2.13 (1.32–3.42)	n/d	n/d	0.0014

* according to predictions on the KMplotter server, URL: https://kmplot.com/ (accessed on 1–20 March 2023) using pan-cancer RNAseq dataset; ** upper quartile survival; PAAD—pancreatic adenocarcinoma, READ—rectum adenocarcinoma, UCEC—uterine corpus endometrioid carcinoma.

**Table 2 ijms-24-10190-t002:** Up-stream regulation of expression patterns of ORF genes, which are critical for cell viability.

Genes *	Up-Regulation of Gene Expression in Cancers	Protein Kinase ** Perturbation Correlates with up-Regulation of ORF Gene Expression	Transcription Factors	Kinase Involvement in Cancer-Associated Signaling Pathways ^#^
Common essential genes ***				
*C1orf109*	THYM	ABL1, ERBB2, FGFR1	CREB1	Pathways in cancer
*C1orf131*	DLBC, THYM	ERBB2, MAPK1	n/d	ErbB receptor tyrosine kinase; gastric cancer, endometrial cancer, lung cancer
*C9orf16*	OV	AKT1, FGFR1	n/d	Pathways in cancer, proteoglycans in cancer
*C12orf45*	DLBC, THYM	AKT1, FGFR2	n/d	Pathways in cancer, PI3K-Akt (phosphoinositide-3-kinase, AKT and mTOR kinases), MAPK (mitogen-activated protein kinase)
*C17orf58*	DLBC, THYM	SYK	n/d	PI3K-Akt
*C19orf53*	DLBC, THYM	ATM, JAK2, MET, PDGFRA	n/d	Pathways in cancer, microRNAs in cancer
Strongly selective genes ***				
*C1orf112*	DLBC, THYM	ABL1, AKT1, JAK1, JAK2	n/d	Pathways in cancer, microRNAs in cancer
*C1orf21*	LAML, PAAD, THYM	AKT1, ERBB3, MAP2K1, MET, TGFBR2	n/d	MAPK, proteoglycans in cancer
*C8orf33*	DLBC, THYM	ABL1, ERBB2, MET, PDGFRA	CREB1	Pathways in cancer, microRNAs in cancer
*C8orf76*	DLBC, THYM	CDK4, MAP2K4, MAPK1	GATA2	Kaposi sarcoma-associated herpesvirus infection, Human T-cell leukemia virus 1 infection
*C9orf40*	DLBC, THYM	ATM, BTK, MET, SYK	n/d	NF-kappa B signaling pathway
*C12orf49*	DLBC, PAAD	ATM, FGFR3	NFYA, USF1	n/d

* Protein-coding genes only; ** genes containing mutations that are causally implicated in cancer (according to the cancer Gene Census); *** according to the DepMap portal; ^#^ pathway mapping was performed at the PathwAX II server, URL: https://pathwax.sbc.su.se/ (accessed on 15–28 February 2023) using the KEGG database v.94.1 as a source. n/d—not determined.

**Table 3 ijms-24-10190-t003:** Potential cancer-associated PPIs involving ORF proteins and CH-associated protein partners.

Protein–Protein Interaction		Gene Co-Expression Data *		The Main Function Related to Protein Partner ***
ORF-Protein	Protein Partner **	Normal Tissue	Cancer Tissue	
C1orf131	EIF3L	Whole blood (0.68)	DLBC (−0.51)	Initiation of protein synthesis
C1orf131	JUN	Whole blood (−0.07)	DLBC (0.61)	Transcriptional master regulator
C19orf53	APEX1	Whole blood (0.87)	DLBC (−0.66)	Cellular response to oxidative stress
C1orf226	BCR	Brain (0.53–0.59)	LGG (−0.06)	Regulatory activities toward small GTP-binding proteins
C1orf226	BTRC	Brain (0.60–0.76)	LGG (−0.09)	Component of E3-ubiquitin-protein ligase complex
C1orf226	PTPRJ	Brain (0.53–0.80)	LGG (0.06)	Dephosphorylation
C2orf88	PRKAR1A	Brain (0.61–0.85)	LGG (0.03)	Regulatory subunit of the cAMP-dependent protein kinases
C6orf132	CDH1	Kidney (0.82)	KIRP (0.04)	Cell adhesion
C1orf131	PPP2R1A	Whole blood (0.74)	THYM (−0.6)	Phosphorylation
C1orf198	ARHGEF1	Whole blood (0.61)	THYM (−0.61)	GTPase-activating protein
C5orf24	PBX1	Whole blood (0.04)	THYM (0.7)	Transcription factor
C11orf98	WWP2	Whole blood (0.75)	THYM (−0.55)	Ubiquitination
C12orf45	SMARCAD1	Whole blood (0.73)	THYM (−0.58)	DNA helicase activity
C12orf45	KPNA1	Whole blood (0.57)	THYM (−0.50)	Nuclear protein import
C1orf115	MDM2	Testis (0.51)	TGCT (−0.41)	Ubiquitination of p53/TP53
C2orf16	SPAG5	Testis (0.68)	TGCT (−0.24)	Chromosome segregation and progression into anaphase
C2orf88	FBXW8	Testis (−0.68)	TGCT (0.54)	Component of ubiquitin-protein ligase complex
C2orf88	GPR161	Testis (−0.64)	TGCT (0.58)	Regulation of Shh signaling
C2orf88	PRKACA	Testis (−0.10)	TGCT (0.53)	Phosphorylation
C6orf132	EZR	Testis (−0.28)	TGCT (0.71)	Connections of cytoskeletal structures to the plasma membrane
C8orf48	OPTN	Testis (0.67)	TGCT (−0.06)	Maintenance of the Golgi complex
C10orf67	BCL2L1	Testis (0.63)	TGCT (−0.22)	Potent inhibitor of cell death
C11orf65	CALM3	Testis (0.81)	TGCT (−0.06)	Calcium signal transduction pathway
C16orf71	MEIS2	Testis (−0.41)	TGCT (0.52)	Transcriptional regulation
C16orf90	RNF123	Testis (0.55)	TGCT (−0.23)	Ubiquitination
C16orf90	NDFIP2	Testis (0.65)	TGCT (0.03)	Activation of HECT domain-containing E3-ubiquitin-protein ligases
C17orf47	ADRM1	Testis (0.65)	TGCT (−0.12)	ATP-dependent degradation of ubiquitinated proteins
C17orf47	NUP214	Testis (0.80)	TGCT (−0.01)	Nuclear pore formation
C19orf44	GAPDHS	Testis (0.83)	TGCT (−0.08)	Regulation of the switch between different pathways for energy production
C6orf132	CDH1 (P12830)	Uterus (0.08)	UCS (0.76)	Calcium-dependent cell adhesion

* Uniprot ID, URL: https://www.uniprot.org/ (accessed on 1 April 2023), ** Spearman correlation coefficient according to the GEPIA2 server, URL: http://gepia2.cancer-pku.cn/#correlation (accessed on 1 April 2023), *** UniprotKB, URL: https://www.uniprot.org (accessed on 1 April 2023). Uniprot IDs of protein partners of ORF proteins: ADRM1 (Q16186), APEX1 (P27695), ARHGEF1 (Q92888), BCL2L1 (Q07817), BCR (P11274), BTRC (Q9Y297), CALM3 (P0DP25), CDH1 (P12830), EIF3L (Q9Y262), EZR (P15311), FBXW8 (Q8N3Y1), GAPDHS (O14556), GPR161 (Q8N6U8), JUN (P05412), KPNA1 (P52294), MDM2 (Q00987), MEIS2 (O14770), NDFIP2 (Q9NV92), NUP214 (P35658), OPTN (Q96CV9), PBX1 (P40424), PPP2R1A (P30153), PRKACA (P17612), PRKAR1A (P10644), PTPRJ (Q12913), RNF123 (Q5XPI4), SMARCAD1 (Q9H4L7), SPAG5 (Q96R06) and WWP2 (O00308).

**Table 4 ijms-24-10190-t004:** Relevant literature data on the known functional characterization of differentially expressed ORF genes in cancers.

Gene	Thesis	Reference
*C1orf54*	*C1orf54* gene-knockout mice exhibit impaired tubular epithelial cell proliferation and delayed recovery after kidney ischemia-reperfusion injury, which lead to deteriorated renal function and increased mortality.	[45]
*C1orf109*	*C1orf109* gene controls the late step of human pre-60S maturation in the cytoplasm and the loss of *C1orf109* global protein synthesis.	[46]
*C1orf115*	C1orf115 protein modulates drug efflux through regulation of the major drug exporter ABCB1/MDR1 by means of physical association with it.	[47]
	CRISPR-Cas9 knockout screens reveal that the loss of *RDD1* (*C1orf115*) resulted in resistance to five anti-cancer drugs.	[48]
*C1orf131*	A protein, encoded by *C1orf131* gene, probably takes part in regulation of rRNA turnover.	[49]
*C1orf194*	The loss of normal *C1orf194* protein altered intracellular Ca^2+^ homeostasis and up-regulated Ca^2+^ handling regulatory proteins.	[50]
*C3orf70*	*C3orf70* gene is involved in neural and neurobehavioral development; defects in *C3orf70* may be associated with neurodevelopmental and neuropsychiatric disorders.	[51]
*C5orf51*	C5orf51 protein interacts with C5orf51 interacts with MON1 and CCZ1, members of the RAB7A guanine nucleotide exchange factor (GEF) complex. C5orf51 is involved in mitophagy (selective degradation of mitochondria by autophagy).	[52]
*C6orf120*	*C6orf120* gene induces apoptosis of CD4^+^ T-cells mediated with endoplasmic reticulum stress	[53]
	Rats with deficiency of *C6orf120* gene are susceptible to liver injury induced by tetra-chloromethane.	[54]
*C8orf76*	C8orf76 protein binds to the promoter region of *SLC7A11* gene and up-regulates *SLC7A11*. *C8orf76* down-regulation induces G1-S arrest and inhibition of cell proliferation.	[55]
*C6orf203*	Knockout of *C6orf203* leads to a decrease in mitochondrial translation and consequent OXPHOS deficiency. C6orf203 involves translation via RNA binding without changing the stability of mtRNAs.	[56,57]
*C11orf42*	C11orf42 protein could potentially be related to trafficking, as it is a part of a complex with 30% similarity to the retromer complex (SNX1, SNX2, VPS35, VPS29 and VPS26A), i.e., with 0.3 Jaccard similarity to the known CORUM retromer complex.	[58]
*C11orf53*	C11orf53 protein interacts with POU2F3 and depletion of C11orf53 reduced enhancer H3K27ac levels and chromatin accessibility, resulting in a reduction of POU2F3-dependent gene expression.	[59,60]
*C11orf88*	C11orf88 protein is required for motile ciliogenesis and flagellar genesis in vertebrates by mediating the maturation of glycolytic enzyme ENO4	[61]
*C11orf94*	Deletion of *C11orf94* gene dramatically decreases male fertility in mice and plays a critical role in sperm binding to the oolemma.	[62]
	Loss of *C11orf94* gene results in reduces assembly of Izumo1 complexes and male infertility due to impaired gamete fusion.	[63]
*C12orf4*	C12orf4 protein is a cytoplasmic protein implicated in the early signaling events following FcεRI-induced cell activation (FcεRI—the high-affinity IgE receptor).	[64]
	From strong in silico evidence, it follows that C12orf4 protein, as a putative ADPR “eraser,” regulates PARP-mediated ADP-ribosylation signaling.	[65]
*C12orf45*	C12orf45 protein acts as a bridge between NOP58 ribonucleoprotein and the particle for arrangement of quaternary structure (PAQosome), a multi-subunit chaperone complex. C12orf45 interacts with NOP58 and RUVBL1/2 AAA^+^ ATPases.	[66]
*C12orf49*	*C12orf49* gene participates in regulation of exogenous lipid uptake through modulation of SREBF2 signaling in response to lipid starvation.	[67]
	C12orf49 protein mediates site-1 protease activation, which is essential for the cleavages of protease substrates in lipid homeostasis.	[68]
*C12orf73*	C12orf73 protein interacts with newly synthesized cytochrome b to support initial steps of complex III biogenesis in complex with UQCC1 and UQCC2.	[69]
*C16orf62*	Heterotrimer (DSCR3, C16orf62 and VPS29), interacting with other protein complexes, takes part in prevention of lysosomal degradation and promotion of cell surface recycling of α5β1 integrin.	[35]
*C16orf71*	Complex C16orf71/Daap1 is identified as a novel axonemal dynein regulator that is critical for ciliary motility.	[19]

## Data Availability

Data on subinteractomes of uncharacterized chromosome-specific open reading frame genes proteins is deposited in doi:10.6084/m9.figshare.22677442.

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
