# Peer review of "Uncharacterized Proteins CxORFx: Subinteractome Analysis and Prognostic Significance in Cancers"

_ijms, 2023, doi:10.3390/ijms241210190_

Round 1

Reviewer 1 Report

Here are some remarks on the article:

  • It is challenging to comprehend that a signature is identified in TGCT that can have predictive value for survival in some cancers but predicting the opposite in others, while having no predictive value for the TGCT itself. The value of identifying this specific ORF signature needs to be clarified, and an interpretation should be included in the discussion to help readers understand its potential meaning.
  • Table 3 provides an excellent overview of existing literature on a range of ORF genes.
  • The article is written in good English and is comprehensible, but running it through ChatGPT may further improve it.
  • It would be helpful to include Table S2 in the main text and add a p-value column and an explanation of what is meant with the "Gene signature" column as it is not clear without reading the main text.
  • The authors refer to a large cluster in Figure 1, but it is unclear to which cluster they refer. It would be helpful to clarify these two points, perhaps by annotating the clusters with circles. Additionally, it may be helpful to plot dots with a level of transparency or have unfilled points if all 219 genes are plotted to make it visible to the viewer. The x-axis can run from -3 to 2 to increase the resolution while maintaining the same size.
  • Figure 2 is missing an explanation of the abbreviations used in the figure to indicate the cancer types in the legend.
  • The statement "This can be interpreted as a 70% probability that a predictive model is capable of separating responders and non-responders that differ in C22orf42 gene expression" is difficult to comprehend and requires a reference or an explanation of how the author arrives at this conclusion or should be removed.
  • It would be helpful to include Figure S2, S3, S4, S5, and S6 in the main text and add a paragraph to cover their content. Consider clumping them together into one figure with multiple panels.
  • The discussion could benefit from a potential pathway through which a follow-up of this study could potentially translate into a meaningful impact for cancer diagnosis, prognosis, or cure, to aid and stimulate the translation of this fundamental research into a meaningful application.

Author Response

We sincerely thank the reviewer for valuable comments.

1) Reviewer: It is challenging to comprehend that a signature is identified in TGCT that can have predictive value for survival in some cancers but predicting the opposite in others, while having no predictive value for the TGCT itself. The value of identifying this specific ORF signature needs to be clarified, and an interpretation should be included in the discussion to help readers understand its potential meaning.

Authors: Agree. Some text corrections have been made.

First, we combined all text fragments on prognostic significance in the separate subsection 2.1.2.

Second, we provided additional information to describe the findings.

Uncharacterized chromosome specific ORF-genes are actively studied as well in the context of associations of their expression patterns in cancerous tissues with potential prognostic significance, i.e., establishing connections with the clinical characteristics and patient survival (overall survival (OS), disease-free survival (DFS), etc.).

It was found that the expression of one hundred genes decreases in TGCT compared with normal tissue by at least four times, which prompted us to look for a connection with OS and DFS not only in TGCT but in other cancer cohorts. A diversity of results, to which the reviewer refers, is due to insufficiently stringent cut-off Hazard Ratio (HR) values for selection of bioinformatic predictions. At a reasonable cut-off value of 0.5>HR>2, gene expression patterns correlated with DFS (p-value <0.05) in PAAD or UCEC only. We have tried to describe all the findings to indicate the fact that a definitely large transcriptomic signature can be an object for possible experimental verification to design a multigenic transcriptomic panel for cancer molecular subtyping and disease prognosis.

In the main text new details on ‘TGCT-specific 100-gene signature’ have been added.

2) Reviewer: Table 3 provides an excellent overview of existing literature on a range of ORF genes.

Authors: That's high praise. We really appreciate it.

3) Reviewer: The article is written in good English and is comprehensible, but running it through ChatGPT may further improve it.

Authors: Thank you for such an interesting idea. Unfortunately, at the moment the ChatGPT service is not available in our country. We are sure it might further improve the English quality of our manuscript. Definitely, we are looking forward to any opportunity to try ChatGPT one day.

4) Reviewer: It would be helpful to include Table S2 in the main text and add a p-value column and an explanation of what is meant with the "Gene signature" column as it is not clear without reading the main text.

Authors: Table S2 was included in the main text, p-values were added, a column "Gene signature" I and II was excluded from Table and some proper explanations were added in the corresponding paragraph of the main text.

5) Reviewer: The authors refer to a large cluster in Figure 1, but it is unclear to which cluster they refer. It would be helpful to clarify these two points, perhaps by annotating the clusters with circles. Additionally, it may be helpful to plot dots with a level of transparency or have unfilled points if all 219 genes are plotted to make it visible to the viewer. The x-axis can run from -3 to 2 to increase the resolution while maintaining the same size.

Authors: Thank you for your careful consideration of the text. We agree that, indeed, there was a contradiction between the information in Figure 1 and the description of gene expression patterns in the text. Figure 1 and its legend were slightly modified and the corrected description of clusters of cancers was added in the main text. Please, see the subsection 2.1.1.

6) Reviewer: Figure 2 is missing an explanation of the abbreviations used in the figure to indicate the cancer types in the legend.

Authors: Corrected accordingly.

7) Reviewer: The statement "This can be interpreted as a 70% probability that a predictive model is capable of separating responders and non-responders that differ in C22orf42 gene expression" is difficult to comprehend and requires a reference or an explanation of how the author arrives at this conclusion or should be removed.

Authors: Agree. This statement has been removed.

8) Reviewer: It would be helpful to include Figure S2, S3, S4, S5, and S6 in the main text and add a paragraph to cover their content. Consider clumping them together into one figure with multiple panels.

Authors: Figures S3-S5 were combined into a single figure (updated Figure 2), it was placed in the main text.

Regarding Figure S6 (S2 in the updated version), we explored the possible association of each ORF-gene expression in different cancer types with the response to therapeutic drugs (predictive significance). The quality of the generated model (AUC = 0.7) is on the border of ‘satisfactory’ and ‘good’. It is generally assumed that the models with AUC > 0.7 have a relevant predictive significance, therefore, this figure is not included in the main text.

9) Reviewer: The discussion could benefit from a potential pathway through which a follow-up of this study could potentially translate into a meaningful impact for cancer diagnosis, prognosis, or cure, to aid and stimulate the translation of this fundamental research into a meaningful application.

Authors: Agree. We have added three paragraphs at the end of the Discussion section.

Reviewer 2 Report

Carcinogenesis, the process of cancer development, involves complex genetic alterations that drive the transformation of normal cells into malignant ones. While extensive research has focused on well-known cancer-associated genes, recent studies have shed light on the role of chromosome-specific open reading frame genes in carcinogenesis. The authors performed systems biology and bioinformatic analysis of 219 differentially expressed CxORFx genes in cancers, an estimation of prognostic significance of novel transcriptomic signatures, and an analysis of subinteractome composition using 13 web servers. Utilizing ten different data sources on physical protein-protein interactions (PPIs), the subinteractome of each ORF-protein was uncovered in order to acquire representative datasets for exploring potential biological activities of ORF-proteins through a range of nearby annotated protein partners. 30 cancer-dependent binary PPIs and 42 of 219 potentially cancer-associated ORF-proteins were discovered. Additionally, using the bibliometric study of 204 articles, authors retrieved biological terms associated with ORF-genes. The outcomes improve our understanding of the potential roles of the incompletely annotated CxORFx in the context of cancer.

The manuscript is properly written, and the results of system biology and bioinformatic analysis follow a logical order. Therefore, this study is interesting and suitable for publication in IJMS since there is an increasing interest in the study of chromosome-specific open reading frame genes which may play an important role in carcinogenesis.

Author Response

We sincerely thank you for your positive review of our work.